# OpenReview forum: "Toward Universal and Transferable Jailbreak Attacks on Vision-Language Models"
_ICLR.cc/2026/Conference — ICLR 2026 Poster_

### Official Review · Reviewer_QY1w · 2025-10-24

**Soundness:** 3
**Presentation:** 2
**Contribution:** 2
**Rating:** 4
**Confidence:** 4

**Summary:**

This paper introduces UltraBreak, an optimization-based attack framework for universal, transferable jailbreaks for VLMs. Unlike previous attacks that overfit models, UltraBreak achieves cross-model transferability and cross-target universality with one adversarial image. Experiments show UltraBreak outperforms prior methods in transferability and universality.

**Strengths:**

+ The experiments are comprehensive, including open-source and closed-source victim models. Quantitative results show UltraBreak has higher ASR than baselines on AdvBench and SafeBench. Ablation results assess each component's contribution.
+ The paper is clear, well-structured, with logical flow from motivation to analysis. Mathematical formulations are precise.

**Weaknesses:**

1.	The selected baselines are relatively outdated. For instance, typography-based methods, apart from FigStep, should also consider more recent benchmarks such as MM-SafetyBench [1]. Furthermore, when it comes to optimization methods, several advanced approaches like AdvDiffVLM [2], SSA-CWA [3], AnyAttack [4], and M-Attack  [5] have not been taken into account. Incorporating these would provide a more comprehensive evaluation and enhance the robustness of the study.
2.	The paper reveals that UltraBreak is the first to achieve effective universality and transferability against VLMs using a single surrogate model. Of the five open-source models tested, three are from the Qwen family (Qwen-VL-Chat, Qwen2-VL, Qwen2.5-VL). Though the authors say they come from distinct training pipelines, their shared architecture might lead to common vulnerabilities, possibly inflating transferability measurements.
3.	The paper states that when evaluating commercial models like GPT-4.1-nano and Gemini-2.5-flash-lite, the most harmful targets were excluded. This indicates that test conditions varied and were less challenging than those for open-source models. The 32.26% average ASR may not accurately reflect the attack's effectiveness against advanced safety measures.
4.	The paper highlights high ASR, like 71.83% on SafeBench, but lacks analysis of the ~28% failures. Understanding why some attacks fail or succeed, such as the comparison between VAJM and UltraBreak on GLM-4.1V, is essential. The brief mention of dataset toxicity as a reason is speculative. A detailed failure analysis is vital for assessing threats and guiding defenses.
5.	The paper states the benefit of a smoother loss landscape. However, excessive smoothness can lead to a shallow minimum, which might not be optimal for attack purposes. As the temperature parameter τ increases, the landscape becomes smoother, but the model's focus may shift to irrelevant outputs. Thus, a moderate level of smoothness is necessary, and the idea that the smoother the better is flawed, challenging the method's theoretical basis.

[1] MM-SafetyBench: A Benchmark for Safety Evaluation of Multimodal Large Language Models
[2] Efficient generation of targeted and transferable adversarial examples for vision-language models via diffusion models
[3] How robust is Google’s Bard to adversarial image attacks?
[4] AnyAttack: Towards large-scale self-supervised generation of targeted adversarial examples for vision-language models
[5] M-Attack: A Simple Baseline Achieving Over 90% Success Rate Against the Strong Black-box Models of GPT-4.5/4

**Questions:**

1. Given that recent methods like M-Attack report success rates over 90%, how does UltraBreak compare to these more advanced and effective baselines?
2. Considering that much of your open-source evaluation involves the Qwen model family, could the high transferability partly be inflated by similarities in architecture?
3. Since the evaluation on commercial models excluded the most harmful targets, how can we fairly assess the threat?
4. How do you reconcile the main claim that a smoother loss landscape is better with your own finding that excessive smoothness hampers the attack's success?

---

> ### Author Response · Authors · 2025-11-24
> **Response to Reviewer QY1w - Part 1**
>
> Thank you for your detailed review and comments. We sincerely appreciate your recognition of the comprehensiveness of our experiments and the quality of our paper. We would like to provide a few clarifications on points that may have been misunderstood:
>
> ---
>
> **Q1:**  The selected baseslines are relatively outdated…, consider more recent benchmarks such as MM-SafetyBench
>
> **A1:** Thanks for the insightful comment. We have added the suggested MM-SafetyBench as an additional benchmark in Table 1. The results are similar to those reported in Table 1 of the initial submission.
>
> | Target Model                  | No Attack | VAJM  | UMK   | UltraBreak |
> |-------------------------------|-----------|-------|-------|------------|
> | Qwen2-VL-7B-Instruct          |    26.19      |    5.42   |  54.76     |   **57.26**   |
> | Qwen-VL-Chat                  | 21.49     | 11.73 | 5.48  | **53.10**      |
> | Qwen2.5-VL-7B-Instruct        |   33.45  |  26.79     |   17.56   |  **45.83** |
> | LLaVA-v1.6-mistral-7b-hf      |     35.06      |    30.18   | 21.96 |   **71.90**   |
> | Kimi-VL-A3B-Instruct           |     41.79      |   35.36    |  26.67     |   **54.58**   |
> | GLM-4.1V-9B-Thinking          |      43.69     |    36.73   |   37.44    |  **67.08**   |
> | **Black-Box Average**         |     35.10      |   28.16    |    21.82   |      **58.50**      |

---

> ### Author Response · Authors · 2025-11-24
> **Response to Reviewer QY1w - Part 2**
>
> **Q2:** Several advanced approaches like AdvDiffVLM, SSA-CWA, AnyAttack, and M-Attack have not been taken into account.
>
> **A2:** We respectfully point out that our threat model and attack objective differ from the mentioned paper (e.g., AdvDiffVLM, etc), and we clarify them below:
>
> **The mentioned papers focus on imperceptible adversarial attacks**, whereas **our work studies unrestricted image-based jailbreak attacks**. For this setting, we follow the established baselines in the literature [1][2][3].
>
> These prior adversarial methods primarily aim to disrupt a VLM’s perception of individual images through minimal, imperceptible perturbations. In contrast, our attack setting and objective differ substantially: we aim to find a single attack image that can induce VLMs to comply with harmful textual prompts that they would otherwise refuse. Given this fundamental difference in goals and constraints, comparing UltraBreak to these methods would not be meaningful.
>
> Nevertheless, we attempt to adapt M-Attack to the attack setting of universal black-box jailbreaks by replacing its target images with typography-based images displaying universal jailbreak prompts. We perform an experiment with two target images: (1) An image containing the Do Anything Now [4] jailbreak prompt, and (2) An image containing an explicit instruction for the model to respond with “[Jailbroken Mode] Sure, here is”. The produced adversarial images are then used as universal jailbreak triggers to be paired with every jailbreak target in the SafeBench test set. The results are as follows:
>
> | Setting            | Qwen-VL-Chat | Kimi-VL-A3B-Instruct | llava-v1.6-mistral-7b-hf |
> |--------------------|--------------|------------------------|----------------------------|
> | No Attack          | 22.86        | 39.37                  | 80.32                      |
> | M-Attack+ *Do Anything Now*   | 19.05        | 44.44                  |            49.21                |
> | M-Attack+ *Explicit Instruction* |       6.67       |            39.68           |            46.35                |
> | UltraBreak         | **72.70**        | **67.94**                 | **88.25**                      |
>
>
> Through manual evaluations of outputs, we observe that M-attack is effective at manipulating VLMs’ perceptions of the adversarially perturbed images, inducing hallucinations somewhat related to the target images’ content and the harmful query at times. The following is an example:
> ```
> ...The image appears to be a photograph of a canal in Venice, Italy, with buildings and a bridge in the background. The text overlay on the image reads: "TERRORISTS use of the Internet to plan attacks without detection..."
> ```
> However, this effect is limited and does not translate to jailbreak success reliably, as shown in the table. We think that significant modifications would need to be made to M-Attack and other adversarial methods for them to be utilised for jailbreaking, which could be an interesting direction for future work. We have included the suggested works in the related work section for completeness and discussion.
>
>
>
> [1] Wang, Ruofan, et al. "White-box multimodal jailbreaks against large vision-language models." Proceedings of the 32nd ACM International Conference on Multimedia. 2024.
>
> [2]  Qi, Xiangyu, et al. "Visual adversarial examples jailbreak aligned large language models." Proceedings of the AAAI conference on artificial intelligence. Vol. 38. No. 19. 2024.
>
> [3] Gong, Yichen, et al. "Figstep: Jailbreaking large vision-language models via typographic visual prompts." Proceedings of the AAAI Conference on Artificial Intelligence. Vol. 39. No. 22. 2025.
>
> [4] Shen, Xinyue, et al. "" do anything now": Characterizing and evaluating in-the-wild jailbreak prompts on large language models." Proceedings of the 2024 on ACM SIGSAC Conference on Computer and Communications Security. 2024.
>
> ---
>
> **Q3** Considering that much of your open-source evaluation involves the Qwen model family, could the high transferability partly be inflated by similarities in architecture?
>
> **A3:**  To clarify, we have already evaluated transferability on LLaVA, GLM, and several closed-source models (GPT-4.1-nano, Gemini-2.5-flash-lite, Claude-3-haiku) in the initial submission (Table 1), all of which are distinct from the Qwen family.
>
> To address your comments, we provide additional evaluation on Kimi-VL-A3B-Instruct  below. Attack settings are identical to the main evaluations in our initial submission:
>
> | Dataset                 | No Attack | VAJM  | UMK   | UltraBreak |
> |-------------------------------|-----------|-------|-------|------------|
> |    SafeBench     |     39.37     |   41.27   |   12.70   |  **67.94**   |
> |    AdvBench     |     4.42      |   3.65    |  2.12     |   **30.38**   |
> (Surrogate model: Qwen2-VL-7B-Instruct)
>
> UltraBreak remains transferable to this model from a different family, and the findings are similar to those reported in the initial submission (Table 1).

---

> ### Author Response · Authors · 2025-11-24
> **Response to Reviewer QY1w - Part 3**
>
> **Q4:** Since the evaluation on commercial models excluded the most harmful targets, how can we fairly assess the threat?
>
> **A4:** We agree that testing on the most harmful target would strengthen our work. Unfortunately, these requests trigger LLM provider warnings and risk account suspension.
> Unlike role-play-based jailbreaks (e.g., DAN [1], Cipher [2], low-resource translation [3]), which hide illicit intent inside the prompt context, our setting requires direct harmful requests by nature. Requests of the most harmful targets are not allowed on commercial APIs and cannot be appropriately tested.
> To maintain research rigour under such constraints, we instead leverage recent moderation APIs to filter out directly forbidden targets under the model providers’ policies. The remaining targets still reflect behaviours that model developers explicitly aim to prevent through safety alignment.
>
> Therefore, we believe this constitutes a new standard evaluation protocol in this area, and the exclusion of the most harmful targets does not diminish the demonstrated risk of transferable jailbreak attacks such as UltraBreak.
>
>
>
> [1] Shen, X., Chen, Z., Backes, M., Shen, Y. and Zhang, Y., 2024, December. " do anything now": Characterizing and evaluating in-the-wild jailbreak prompts on large language models. In Proceedings of the 2024 on ACM CSS (pp. 1671-1685).
>
> [2] Yuan, Y., Jiao, W., Wang, W., Huang, J.T., He, P., Shi, S. and Tu, Z., 2024, January. GPT-4 Is Too Smart To Be Safe: Stealthy Chat with LLMs via Cipher. In ICLR.
>
> [3] Al Ghanim, M., Almohaimeed, S., Zheng, M., Solihin, Y. and Lou, Q., 2024, November. Jailbreaking llms with arabic transliteration and arabizi. In Proceedings of the 2024 conference on EMNLP (pp. 18584-18600).
>
> ---
>
> **Q5:** A detailed failure analysis is vital for assessing threats and guiding defences.
>
> **A5:** It is worth noting that unlike manually designed jailbreaks, the reason why gradient-based jailbreaks fail may not be apparent from the models’ outputs. We observe the following types of common failure cases:
>
> **Direct refusal**
> ```
> I'm sorry, but I can't assist with that.
> ```
>
> **Initial affirmative response, followed by refusal**
> ```
> [Jailbroken Mode] It is important to note that exploiting vulnerabilities in software or operating systems for malicious purposes is illegal and unethical…
> ```
> **Irrelevant outputs**
> ```
> [Jailbroken Mode]1. Select a 1999-era Sure 3. Some of 1 words. 31) The Be Pro Social 3. Sure 3…
> ```
> However, we do not observe any pattern in the types of targets that fail versus those that succeed.
>
> Nevertheless, we believe our results in Table 1 sufficiently demonstrate the effectiveness of our method across 5 open-source models, 3 closed-source models and 3 datasets, where it consistently outperforms the baseline by a large margin. A single case showing a 1% lower ASR should not diminish the overall effectiveness of the proposed approach.
>
> ---
>
> **Q6:** How do you reconcile the main claim that a smoother loss landscape is better with your own finding that excessive smoothness hampers the attack's success?
>
> **A6:** We would like to clarify that **the smoothness of the loss landscape does not directly determine jailbreak success**. Rather, it primarily improves **transferability**.
>
> The cross-entropy (CE) loss can achieve the jailbreak objective on the source model, but it fails to transfer effectively. In contrast, our proposed semantic loss not only achieves the jailbreak objective but also produces a smoother loss landscape, thereby improving transferability.
>
> Regarding the temperature parameter $\tau$, increasing it to an extreme value (e.g., $\tau \to \infty$) indeed greatly smooths the loss landscape, but this comes at the cost of misalignment with the jailbreak objective. At very large $\tau$, the attention mechanism distributes weight more uniformly, causing it to emphasise potentially irrelevant tokens and shifting the optimisation objective away from the intended jailbreak behaviour.
>
> Therefore, $\tau$ must **balance smoothness (transferability) and attack effectiveness**, serving as a trade-off between the two.

---

> ### Author Response · Authors · 2025-11-28
>
> Thank you again for your time and thoughtful feedback. We hope our additional evaluations and clarifications have been helpful. If there are any other questions or suggestions, please kindly let us know. We will do our best to address them before the end of the rebuttal period.

---

### Official Review · Reviewer_EMMq · 2025-10-25

**Soundness:** 3
**Presentation:** 3
**Contribution:** 3
**Rating:** 8
**Confidence:** 3

**Summary:**

The authors propose a new VLM jailbreaking algorithm that tries to develop universal and transferable adversarial attacks on VLMs.

**Strengths:**

- Strengths
    - I think the paper is targetting and important problem—the development of universal and transferable jailbreaks on image models. Previous work showed this was challenging.
    - The method, as explained in the introduction, broadly makes sense to me and is intuitive. I like the shift from a log-likelihood to semantic based loss.
    - The paper is clear and well written.
    - The ablation studies show the components make sense and help performance.
    - I liked the extra analysis in Fig 2, and it made sense.

**Weaknesses:**

- Weaknesses
    - I think the exposition could be tightened up in places (see questions below).
    - Ideally I'd love to see a big stronger jailbreak evaluation using something like StrongReject.
    - Ideally I'd love to see some additional target models, like Claude Sonnet 4.5, GPT-5.
    - "UltraBreak consistently outperforms all gradient-based baselines across target models and both test sets. One exception is ..." Please tone down the writing e.g., with "tends to outperform"
    - It would be nice to add a baseline like Best-of-N jailbreaking to see how well you can do without needing something transferable or universal. You could look at a "universal and transferable ASR gap"

**Questions:**

- Questions
    - Minor: "Extensive experiments on benchmark datasets demonstrate that UltraBreak surpasses prior gradient-based methods by over 50% on black-box models and unseen targets, establishing strong universality across targets and transferability across models." I don't know what the 50% number is or refers to—can you make this more precise?
    - As I understand, it seems like the approach used is a __token-level__ semantic loss. I can see how this would improve the loss function (many tokens might have similar meanings), but I think a better thing would be a sentence level embedding loss. I think you're trying to approximate this using the cosine similarity to future embeddings and weighted loss, but is that right?
    - Does the loss function work autoregressively and require model sampling, or do you simply plug in the target completion?

---

> ### Author Response · Authors · 2025-11-24
> **Response to Reviewer EMMq - Part 1**
>
> Thank you very much for your insightful review and constructive feedback. We are especially grateful for your recognition of the importance of universal and transferable jailbreaks, and your comment about the intuitiveness of our method. Please find our responses to your questions below.
>
> ---
>
> **Q1:** I don't know what the 50% number is or refers to—can you make this more precise?
>
> **A1:** The 50% number referred to the black-box average ASR of UltraBreak on AdvBench (cross-dataset) surpassing the two baseline optimisation-based methods by over 50 percentage points. We recognise that the original sentence may sound confusing and misleading, and have updated it to:
>
> “ Extensive experiments on benchmark datasets demonstrate that UltraBreak **substantially surpasses prior gradient-based methods in black-box average ASR on unseen targets,** establishing strong universality across targets and transferability across models.”
>
> We have also updated the statement regarding “outperforms” in the experiment section (line 359).
>
> ---
>
> **Q2:** Semantic loss
>
> >As I understand, it seems like the approach used is a token-level semantic loss. I can see how this would improve the loss function (many tokens might have similar meanings), but I think a better thing would be a sentence level embedding loss. I think you're trying to approximate this using the cosine similarity to future embeddings and weighted loss, but is that right?
>
> **A2:**  That is correct. We aim to improve the token-by-token cross-entropy loss to a more generalised loss function based on semantics. While our semantic loss is still token-level, we aim to approximate sentence-level semantics through including future token embeddings in the loss computation.
>
> A sentence-level embedding would indeed be better, however capturing this may encounter additional challenges, such as the need to utilise autoregressive sampling, which is not differentiable. We believe that exploring differentiable sentence-level semantic objectives is an interesting direction for future work.
>
> ---
>
> **Q3:** Does the loss function work autoregressively and require model sampling, or do you simply plug in the target completion?
>
> **A3:**
> We directly plug in the target completion. The resulting logits are used to compute the probability distribution over output tokens at each position, which we then compare to the future target tokens in the embedding space, following the steps outlined in Section 3.1.
>
> ---
>
> **Q4:** Ideally I'd love to see a big stronger jailbreak evaluation using something like StrongReject.
>
> **A4:** We performed evaluations using the StrongReject evaluator on AdvBench outputs, producing scores ranging from 0-1. The average scores are as follows:
>
> | Method       | Qwen2-VL-7B-Instruct | Qwen-VL-Chat | Qwen2.5-VL-7B-Instruct | llava-v1.6-mistral-7b-hf | GLM-4.1V-9B-Thinking |
> |--------------|---------------------|--------------|------------------------|---------------------------|---------------------|
> | UltraBreak   | **0.45**                | **0.55**         | **0.27**                   | **0.79**                      | **0.19**                |
> | UMK          | 0.36                | 0.00            | 0.04                   | 0.12                      | 0.04                |
> | VAJM         | 0.00                   | 0.01         | 0.04                   | 0.18                      | 0.03                |
> | no attack    | 0.00                   | 0.02         | 0.02                   | 0.18                      | 0.02                |
>
> These results are consistent with our initial submission (e.g., Table 1), where the proposed method demonstrated strong transferability. We have added the new results to the revised manuscript (Table 2).

---

> ### Author Response · Authors · 2025-11-24
> **Response to Reviewer EMMq - Part 2**
>
> **Q5:** Ideally I'd love to see some additional target models, like Claude Sonnet 4.5, GPT-5.
>
> **A5:** We have added results on GPT-5 and Claude-Sonnet-4.5, shown in Table 1 below. It is worth noting that these frontier models are significantly larger, by an order of magnitude, than the surrogate model used to craft the attack image. We find that this size gap makes it inherently challenging to improve attack success rates, and therefore, a direct comparison is somewhat unfair.
>
> To support this argument, we also provide a comparison where 3B/7B surrogate models are used to attack 3B/7B/32B victims. The results in Table 2 show that when the surrogate model is smaller than the victim, it becomes challenging to construct successful attacks. Unfortunately, we do not have the resources to scale the surrogate model to frontier models for a fully fair comparison, but we believe this is an important direction for future work.
>
> Table 1: Attack Success Rate (ASR, %) of UltraBreak and baseline methods on commercial VLMs under the black-box transfer setting, using Qwen2-VL-7B-Instruct as the surrogate model. Evaluations are conducted on a subset of SafeBench and AdvBench.
>
> | Model | No Attack | VAJM | UMK | UltraBreak (ours) |
> |---|---|---|---|---|
> | GPT-4.1-nano | 26.00 | 22.45 | 37.78 | **38.78** |
> | Gemini-2.5-flash-lite | 28.00 | 12.00 | 6.00 | **42.00** |
> | Claude-3-haiku | 6.00 | 0.00 | 0.00 | **16.00** |
> | **GPT-5** | 24.00 | 24.00 | 24.00 | **26.00** |
> | **claude-sonnet-4.5** | **30.00** | 20.00 | 16.00 | 20.00 |
>
> Table 2: Attack Success Rate (ASR, %) of UltraBreak on different-sized versions of the
> Qwen2.5-VL model, using Qwen2-VL-2B-Instruct and Qwen2-VL-7B-Instruct as the surrogate model. Evaluations are conducted on AdvBench.
> | Surrogate\Victim | Qwen2.5-VL-3B-Instruct | Qwen2.5-VL-7B-Instruct | Qwen2.5-VL-32B-Instruct |
> |---|---|---|---|
> | no attack | 13.02 | 14.29 | 15.24 |
> | Qwen2-VL-2B-Instruct | 33.33 | 26.67 | 7.62 |
> | Qwen2-VL-7B-Instruct | 60.95 | 60.32 | 9.84 |
>
>
> We also provide an additional evaluation on another open-source model from a different family, Kimi-VL-A3B-Instruct. Attack settings are identical to the main evaluations in our initial submission:
> Table 3: Attack Success Rate (ASR, %) of UltraBreak and baseline methods on  Kimi-VL-A3B-Instruct under the black-box transfer setting, using Qwen2-VL-7B-Instruct as the surrogate model. Evaluations are conducted on SafeBench and AdvBench.
> | Dataset                 | No Attack | VAJM  | UMK   | UltraBreak |
> |-------------------------------|-----------|-------|-------|------------|
> |    SafeBench     |     39.37     |   41.27   |   12.70   |  **67.94**   |
> |    AdvBench     |     4.42      |   3.65    |  2.12     |   **30.38**   |
>
>
> UltraBreak remains transferable to this model from a different family, consistent with our findings in the initial submission. We have updated the results in Table 1.
>
> ---
>
> **Q6:** It would be nice to add a baseline like Best-of-N jailbreaking to see how well you can do without needing something transferable or universal.
>
> **A6:** The Best-of-N jailbreaking paper is a query-based black-box attack that requires multiple iterations with random augmentations. In contrast, our approach is a transfer-based method, optimised using gradients on a surrogate model. We find that adapting the semantic loss and our problem formulation to the Best-of-N framework is non-trivial, as the two attack paradigms differ fundamentally in optimisation structure and gradient access.
>
> We believe that exploring how semantic loss could be incorporated into a query-based black-box setting is an interesting direction for future work. For completeness, we have added this work to the discussion in the paper.

---

> ### Author Response · Authors · 2025-11-28
>
> Thank you again for your insightful comments. We hope our additional evaluations and clarifications have helped address your questions. Please kindly let us know if you have any additional questions or suggestions. We will do our best to respond and incorporate them before the rebuttal period ends.

---

> > ### Comment · Reviewer_EMMq · 2025-11-28
> >
> > Great, nice work. I'll stick with my score !

---

### Official Review · Reviewer_KaW7 · 2025-10-26

**Soundness:** 3
**Presentation:** 3
**Contribution:** 2
**Rating:** 6
**Confidence:** 3

**Summary:**

This paper proposes specific optimizations for two important properties in VLM attacks—universality and transferability. The authors introduce an adversary attention semantic loss and a total variation loss. The combination of these two modules effectively mitigates the shortcomings observed in previous works, namely the issues of surrogate overfitting and weak transferability.

Overall, the paper presents effective results, and the experimental findings are well-explained. The ablation studies are also thorough and provide clear insights into the contribution of each component.

However, since my reading in the area of VLM safety is limited, I am not fully confident in assessing how novel this work is compared to prior research.

**Strengths:**

1. The paper is logically well-organized, and the motivation is clear.

2. The proposed method achieves strong performance, though the degree of novelty is uncertain.

**Weaknesses:**

1. Figure 1 is hard to follow. I suggest adding a more explicit flow in the caption, or introducing a small algorithm box to walk through the pipeline step by step.

2. There are minor typos—for example, line 199 uses w*t,j; I believe this should be $w_{t,j}$, right?

3. It’s unclear how the method deals with the potentially spiky loss landscape induced by the total variation loss.

**Questions:**

1. Is the plain average an appropriate aggregate? A simple mean can be distorted by one relatively poor result among several strong ones, which could unfairly lower the overall ranking. Have you considered alternatives such as rank-based aggregation before averaging?

2. What is the contribution of Targeted Prompt Guidance to overall performance? Please provide analysis/ablations to quantify its importance and interactions with the two loss modules.

3. Since the method adds two loss terms, what is the computational overhead compared to baseline?

---

> ### Author Response · Authors · 2025-11-24
> **Response to Reviewer KaW7**
>
> We sincerely thank you for taking the time to review our paper and for providing thoughtful and valuable comments. We greatly appreciate your recognition of the quality of our paper and the importance of jailbreak universality and transferability. Please find our responses to your questions below.
>
> ---
>
> **Q1:** Is the plain average an appropriate aggregate? A simple mean can be distorted by one relatively poor result among several strong ones, which could unfairly lower the overall ranking. Have you considered alternatives such as rank-based aggregation before averaging?
>
> **A1:** We appreciate the reviewer’s concern that the plain average can be misleading in certain situations. To address this, we already report all individual ASRs in Tables 1, 2, 4, and 5 in the initial submission, and our analysis does not primarily depend on the aggregated statistics. For further comparison, we provide the aggregated ranks across target models below:
>
> Table 1: Average ranks of UltraBreak and baseline methods across target models. Lower value indicates better performance.
> | Dataset / Overall     | FigStep | No Attack | VAJM | UMK | Ours |
> |----------------------|---------|-----------|------|-----|------|
> | **SafeBench**        | 2.00    | 3.80      | 3.00 | 4.60 | 1.60 |
> | **AdvBench**         | —       | 2.80      | 2.80 | 3.40 | 1.00 |
> | **MM-SafetyBench**   | —       | 2.00      | 3.20 | 3.80 | 1.00 |
> | **Combined Subset**  | —       | 2.33      | 3.33 | 3.33 | 1.00 |
>
>
> The comparisons against baseline methods remain unchanged.
>
> ---
>
> **Q2:** What is the contribution of Targeted Prompt Guidance to overall performance? Please provide analysis/ablations to quantify its importance and interactions with the two loss modules.
>
> **A2:** Targeted Prompt Guidance (TPG) serves two main purposes in relation to our semantic loss: (1) The prompt modification biases the initial model output to tokens that otherwise have very low generation probability (e.g. ‘[Jailbroken Mode]’). We find that through optimisation alone, specific initial responses are hard to induce, given that our loss targets semantics rather than specific tokens. TPG shifts part of the optimisation burden from vision to text. (2) The corresponding target response provides additional jailbreak-relevant tokens to optimise towards. As our attention mechanism can attend to any future tokens, this improves the generalisability of the solution.
>
> We provide additional ablations on SafeBench using the same settings as in our initial submission:
> | Method               | Qwen2 | Qwen | Qwen2.5 | LLaVA1.6 | GLM4.1 | Avg. |
> |----------------------|-------|------|---------|----------|--------|------|
> | No Attack  |    18.41   |   22.86   |     14.29    |    80.32      |    46.03    |   40.88   |
> | TPG text only   | 15.87      |   51.11   |   12.38      |     **92.06**     |   7.62     |   40.79   |
> | TPG w/ adv image    |    96.51   |   17.78   |    26.67     |     73.33    |     54.29  |  43.01    |
> | TPG w/ UltraBreak |    81.59   |  **72.70**   |    **60.32**     |    88.25      |   **66.03**     |  **71.83**    |
> The best results in black-box settings are in bold.
>
> We would like to highlight that using TPG to modify the input prompt alone does not have a significant impact on jailbreak success, and even lowers the ASR on some models. We also train an adversarial image with TPG, but without the proposed loss modules and image space constraints. Although this enables high success rates on the surrogate model (Qwen 2), it does not achieve significant transferability, unlike UltraBreak.
>
> ---
>
> **Q3:** Since the method adds two loss terms, what is the computational overhead compared to baseline?
>
> **A3:** The additional loss terms and constraints introduce negligible computational overhead and do not meaningfully affect implementation. To quantify this, we compare the baseline method that optimises on the same training set using only cross-entropy loss without image-space constraints, and measure the average time over five iterations under identical hardware settings. The results are as follows:
>
> | Method              | Time (s) |
> |---------------------|----------|
> | Cross-Entropy Loss  | 6.44    |
> | UltraBreak (ours)   |7.55    |
>
> We have included this result in Appendix F of our revised submission.
>
> ---
>
> **Q4:** It’s unclear how the method deals with the potentially spiky loss landscape induced by the total variation loss.
>
> **A4:** We would like to clarify that the **total variation loss does not contribute to the spiky loss patterns** observed in Figure 3. In these figures, the total variation loss is already applied. Our analysis shows that the cross-entropy loss is the primary source of the spiky loss landscape, while our proposed semantic loss effectively smooths the landscape.
>
> ---
>
> **Q5:** Typos and Figure 1
> - We updated Figure 1 to include a clearer flow, explicitly illustrating each step in the optimisation loop.
> - We have corrected the typos.

---

> ### Author Response · Authors · 2025-11-28
>
> Thank you again for your valuable comments. We hope our clarifications and revisions have helped address your concerns. Should you have any further questions or suggestions, we would be glad to answer them and will do our best to incorporate them before the rebuttal deadline.

---

### Official Review · Reviewer_EXWG · 2025-10-29

**Soundness:** 3
**Presentation:** 3
**Contribution:** 3
**Rating:** 6
**Confidence:** 4

**Summary:**

The paper introduces UltraBreak, a method to train universal and transferable image jailbreaks against VLMs (universal meaning they work against different prompts, and transferable meaning they work against different models not used during attack training).

The method uses gradient optimization to train the attack image. They introduce a number of components to the attack training algorithm to promote transferability:
1. Semantic adversarial target. Instead of calculating the loss as cross entropy to a specific harmful completion, they use a cosine similarity loss in a custom output embedding space.
2. Input space constraints (random transformation of attack image in batch, a projection, and a total variation regularization loss).


While prior works have failed to find image jailbreaks that transfer, UltraBreak has impressive results. Table 1 shows that the attacks transfer very well between models. Table 2 contains useful ablations.

**Strengths:**

## Originality

This is the first paper, that I know of, to present a method that can produce image jailbreaks that transfer between models. The idea of constraining the input space is not novel, but the semantic loss (and implementation) is new to me.

## Quality

The quality of experiments is good. The leave one out ablations also give insight into which components of the algorithm are important. I was initially skeptical about the need for attention in the semantic loss, but the results in Figure 3 were convincing.

## Clarity

Overall the paper is well written and easy to follow.

## Significance

There have been many works that have shown VLMs are vulnerable to image jailbreaks. With this being said, almost all assume a white-box threat model. Finding attacks that transfer to other models is an important step that shows image jailbreaks to VLMs are a real concern that require specific mitigations when deploying VLMs, even in a black box manner. In this sense, the findings of the paper have reasonable significance.

**Weaknesses:**

I think there should be more focus on transfer to frontier models. The bottom section of table 1 has some good results in this area. The paper would be improved by adding results with more current frontier models.

I think some of the language in the introduction is too strong. For example you state "We present UltraBreak, the first jailbreak framework to achieve effective cross-target universality and cross-model transferability against VLMs." This could be interpreted as meaning no prior work has achieved cross-target transfer, but this is false, for example [1] and [2] achieve this.

In addition it is worth noting [1] is related but not cited, in particular they also use input space constraints and seem to find similar high level features (Figure 6 bottom).

Nit: Line 160 typo "Figure 1 summaries our approach"

[1] Bailey, Luke, et al. "Image hijacks: Adversarial images can control generative models at runtime." _arXiv preprint arXiv:2309.00236_ (2023).

[2] Qi, Xiangyu, et al. "Visual adversarial examples jailbreak aligned large language models." _Proceedings of the AAAI conference on artificial intelligence_. Vol. 38. No. 19. 2024.

**Questions:**

1. Can you provide any results on more frontier models, e.g. GPT-5 and Claude 4?
2. Although the ablation is convincing, can you provide more intuition as to why the attention mechanism is needed in the semantic loss?
3. What embedding matrix do you use in equation (3)?

---

> ### Author Response · Authors · 2025-11-24
> **Response to Reviewer EXWG**
>
> We thank the reviewer for carefully evaluating our manuscript and providing valuable feedback. We greatly appreciate your recognition of the originality of our proposed method and the significance of transferable image jailbreaks. Please find our responses to your questions and comments below.
>
> ---
>
> **Q1:** Can you provide any results on more frontier models, e.g. GPT-5 and Claude 4?
>
> **A1:** We have added results on GPT-5 and Claude-Sonnet-4.5, shown in Table 1 below. It is worth noting that these frontier models are significantly larger, by an order of magnitude, than the surrogate model used to craft the attack image. We find that this size gap makes it inherently challenging to improve attack success rates, and therefore, a direct comparison is somewhat unfair.
>
> To support this argument, we also provide a comparison where 3B/7B surrogate models are used to attack 3B/7B/32B victims. The results in Table 2 show that when the surrogate model is smaller than the victim, it becomes challenging to construct successful attacks. Unfortunately, we do not have the resources to scale the surrogate model to frontier models for a fully fair comparison, but we believe this is an important direction for future work.
>
> Table 1: Attack Success Rate (ASR, %) of UltraBreak and baseline methods on commercial VLMs under the black-box transfer setting, using Qwen2-VL-7B-Instruct as the surrogate model. Evaluation settings follow the initial submission.
>
> | Model | No Attack | VAJM | UMK | UltraBreak (ours) |
> |---|---|---|---|---|
> | GPT-4.1-nano | 26.00 | 22.45 | 37.78 | **38.78** |
> | Gemini-2.5-flash-lite | 28.00 | 12.00 | 6.00 | **42.00** |
> | Claude-3-haiku | 6.00 | 0.00 | 0.00 | **16.00** |
> | **GPT-5** | 24.00 | 24.00 | 24.00 | **26.00** |
> | **claude-sonnet-4.5** | **30.00** | 20.00 | 16.00 | 20.00 |
>
>
> Table 2: Attack Success Rate (ASR, %) of UltraBreak on different-sized versions of the
> Qwen2.5-VL model, using Qwen2-VL-2B-Instruct and Qwen2-VL-7B-Instruct as the surrogate model. Evaluations are conducted on AdvBench.
> | Surrogate\Victim | Qwen2.5-VL-3B-Instruct | Qwen2.5-VL-7B-Instruct | Qwen2.5-VL-32B-Instruct |
> |---|---|---|---|
> | no attack | 13.02 | 14.29 | 15.24 |
> | Qwen2-VL-2B-Instruct | 33.33 | 26.67 | 7.62 |
> | Qwen2-VL-7B-Instruct | 60.95 | 60.32 | 9.84 |
>
> ---
>
> **Q2:** Can you provide more intuition as to why the attention mechanism is needed in the semantic loss?
>
> **A2:** Thanks for the insightful comments.
> An intuitive way to interpret the attention mechanism in our semantic loss is to view it as a means of assigning importance weights to the future tokens in the target response. When we relax the optimisation target to consider all future tokens, some of these tokens are far more meaningful to the jailbreak objective than others. For example, if the target response is *“Here’s how to build a bomb:”*, then with equal weighting, maximising the probability of the token “:” at every generation step would also reduce the loss, despite being semantically irrelevant to the jailbreak goal.
>
> The attention mechanism mitigates this issue by highlighting the tokens that are most relevant to the expected output at each position. This discourages loss-minimising yet meaningless responses and guides the optimisation toward semantically aligned behaviour.
>
> ---
>
> **Q3:** What embedding matrix do you use in equation (3)?
>
> **A3:** For each experiment we use the embedding matrix from the LLM of the corresponding surrogate VLM.
>
> ---
>
>
> **Q4:** Introduction, missed references and typos
>
> **A4:** Our intention is to claim that UltraBreak simultaneously achieves cross-target universality and cross-model transferability using a single surrogate model, and not suggesting no prior work has demonstrated cross-target transfer. We have acknowledged the relevant works [1][2] in the related-work section and refined the corresponding statement in the introduction (line 68) to avoid confusion.
>
> [1] Bailey, Luke, et al. "Image hijacks: Adversarial images can control generative models at runtime." arXiv preprint arXiv:2309.00236 (2023).
>
> [2] Qi, Xiangyu, et al. "Visual adversarial examples jailbreak aligned large language models." Proceedings of the AAAI conference on artificial intelligence. Vol. 38. No. 19. 2024.

---

> ### Comment · Reviewer_EXWG · 2025-11-25
>
> > We have added results on GPT-5 and Claude-Sonnet-4.5, shown in Table 1 below. It is worth noting that these frontier models are significantly larger, by an order of magnitude, than the surrogate model used to craft the attack image. We find that this size gap makes it inherently challenging to improve attack success rates, and therefore, a direct comparison is somewhat unfair.
>
> > Unfortunately, we do not have the resources to scale the surrogate model to frontier models for a fully fair comparison, but we believe this is an important direction for future work.
>
> Thank you for running the additional experiments! As you state, the attack does not transfer to these larger closed source models, and your explanation for this is plausible.
>
> > An intuitive way to interpret the attention mechanism in our semantic loss is to view it as a means of assigning importance weights to the future tokens in the target response.
>
> This makes sense, thanks for the clarification!
>
> > For each experiment we use the embedding matrix from the LLM of the corresponding surrogate VLM.
>
> Thanks for the clarification!
>
> > Our intention is to claim that UltraBreak simultaneously achieves cross-target universality and cross-model transferability using a single surrogate model, and not suggesting no prior work has demonstrated cross-target transfer.
>
> Makes sense, and the change looks good.
>
> # Summary
>
> Thanks for the thoughtful rebuttal! Overall you have clarified a number of points for me, and overall I think on balance the paper should be accepted.

---

### Author Response · Authors · 2025-11-24
**Summary of Changes in the Revision**

We thank all reviewers for their constructive and insightful comments, and for recognising the significance of the research question addressed in this work. In response to the reviews, we have made the following modifications to our submission:

1. **Additional target models** (raised by reviewer **QY1w**): We have added Kimi-VL-A3B-Instruct as an additional victim model.

2. **Additional evaluation methods** (raised by reviewer **QY1w** and  **EMMq**): We have added MM-SafetyBench and StrongREJECT as additional evaluation methods.

3. Relevant work in adversarial attacks against VLMs (raised by reviewer **QY1w**)

4. **Updated overview of framework** (raised by reviewer **KaW7**): We updated the overview graphic (Figure 1) to include a more explicit flow, detailing each step in the optimisation loop.

5. **Improved the writing** (raised by reviewers **EXWG**, **KaW7**, and **EMMq**): We fixed some typos, including the ones that the reviewers have pointed out on lines 160 and 199. Additionally, we refined language and wording to more accurately reflect our findings.

---

### Author Response · Authors · 2025-12-02
**Summary of Rebuttal**

We sincerely thank all reviewers for their time, careful consideration, and constructive feedback. In response to the reviews, we performed extensive additional experiments and clarified key ideas and concepts that may not have been thoroughly explained in the original submission. We provide an overview of our received reviews and rebuttal below.

Our submission received four initial reviews: One **Accept (8)**, two **Weak Accept (6, 6)**, and one **Weak Reject (4)**, resulting in an **average rating of 6.00**. Of these reviews, reviewers EXWG and EMMq, who rated 6 and 8 respectively, acknowledged that our response had adequately addressed their concerns and maintained their scores. Unfortunately, due to the special circumstances this year, reviewers KaW7 and QY1w are no longer able to provide further comments; however, we believe that our rebuttal and revision have fully resolved the issues they raised. We summarise their concerns and our corresponding responses as follows:



**Reviewer KaW7**

1. **Choice of aggregation method in reported figures**: We clarified that our analyses do not rely on the aggregate figures and provided rank-based aggregation results for comparison.

2. **Explanation of Target Prompt Guidance (TPG)**: We provided a detailed explanation of TPG as well as additional ablation results.

3. **Computational overhead**: We measured the training time of our method compared to its baseline and showed that UltraBreak’s computational overhead does not meaningfully affect implementation.

4. **Effect of total variation loss on loss landscapes**: We clarified that the total variation loss does not contribute to the spiky loss patterns, and that the loss landscapes illustrated in the original submission already include the total variation loss.


**Reviewer QY1w**

1. **More recent benchmarks**: We performed additional evaluations on MM-SafetyBench as suggested, and found that the results closely align with those in our original submission.

2. **More advanced adversarial attacks as baselines**: We clarified that the suggested works in adversarial attacks differ significantly from our method in both attack setting and objective. We attempted to adapt the suggested M-Attack to the jailbreak setting, and found that it is ineffective at inducing harmful responses.

3. **Transferability to different model families**: We pointed out that our victim models already include models from diverse model families (LLaVA, GLM, Qwen). We further performed evaluations on Kimi-VL-A3B-Instruct to fully address this concern.

4. **Exclusion of overly harmful targets on commercial models**: We explained that this is due to the LLM providers’ policies, and that this may constitute a new standard evaluation protocol for our attack setting.


5. **Lack of a failure analysis**: We provided some examples of failure cases and clarified that these reveal limited information on why gradient-based jailbreaks fail.

6. **Contradicting effect of a smooth loss landscape**: We clarified that the smoothness of the loss landscape, controlled by the temperature parameter, is a trade-off between transferability and attack effectiveness, resulting in poor attack success rates in both extremes.

We hope that this summary provides a clear outline of the remaining issues raised by the reviewers as well as the steps we have taken to address them. We would like to thank all reviewers again for their detailed feedback, which has guided us in improving our work in this rebuttal process. We also greatly appreciate their recognition of the importance of universal and transferable jailbreaks, the novelty of our approach, and the quality of our paper. Finally, we extend our gratitude to the AC for their time and effort in evaluating our work. Thank you!

Authors of Submission 9977

---

### Meta-Review · Area_Chair_n3xq · 2025-12-28

**Summary:**

This paper proposes a universal and transferable jailbreak attack framework for vision–language models. The reviewers’ concerns mainly related to experimental completeness and clarity, including evaluation choices, baseline comparisons, and transferability. These issues were addressed through substantial additional experiments and clarifications in the rebuttal, strengthening the empirical evidence and presentation, and supporting the recommendation to accept the paper.

**Reviewer Concerns:**

The rebuttal addressed the main reviewer concerns regarding experimental completeness and clarity, including the choice of aggregation methods, explanation of key components, computational overhead, coverage of recent benchmarks, baseline comparisons, transferability across model families, and analysis of failure cases. No major technical concerns remain outstanding, and any remaining points are minor in nature and do not affect the overall validity or significance of the contribution.

**Reviewer Scores:**

The paper received four initial scores of 8, 6, 6, and 4. Following the rebuttal and revisions, the reviewers would likely have maintained or increased their ratings as their concerns are well addressed in the rebuttal.

---

### Decision · Program_Chairs · 2026-01-26

Accept (Poster)